

# Evaluating solar radiation forecast uncertainty

Minttu Tuononen[1], Ewan J. O'Connor[1,2], and Victoria A. Sinclair[3]

[1]Finnish Meteorological Institute, Helsinki, Finland
[2]Department of Meteorology, University of Reading, United Kingdom
[3]Institute for Atmospheric and Earth System Research / Physics, Faculty of Science, University of Helsinki, Helsinki, Finland

**Correspondence:** Minttu Tuononen (minttu.tuononen@fmi.fi)

**Abstract.** The presence of clouds, and their characteristics, has a strong impact on the radiative balance of the Earth and on the amount of solar radiation reaching the Earth's surface. Many applications require accurate forecasts of surface radiation on weather timescales, for example, solar energy and UV radiation forecasts. Here we investigate how operational forecasts of low and mid-level clouds affect the accuracy of solar radiation forecasts. Four years of cloud and solar radiation observations

from one site - Helsinki, Finland, are analysed. Cloud observations are obtained from a ceilometer and therefore, we first develop algorithms to reliably detect cloud base, precipitation and fog. These new algorithms are widely applicable for both operational use and research, such as in-cloud icing detection for the wind energy industry and for aviation. The cloud and radiation observations are compared to forecasts from the Integrated Forecast System (IFS) run operationally and developed by the European Centre for Medium-Range Weather Forecasts (ECMWF). We develop methods to evaluate the skill of the cloud

and radiation forecasts. These methods can potentially be extended to hundreds of sites globally.

   Over Helsinki, the measured Global Horizontal Irradiance (GHI) is strongly influenced by its northerly location and the annual variation in cloudiness. Solar radiation forecast error is therefore larger in summer than in winter, but the relative error in the solar radiation forecast is more or less constant throughout the year. The mean overall bias in the GHI forecast is positive (8 W m$^{-2}$). The observed and forecast distributions in cloud cover, at the spatial scales we are considering, are

strongly skewed towards clear-sky and overcast situations. Cloud cover forecasts show more skill in winter when the cloud cover is predominantly overcast; in summer there are more clear-sky and broken cloud situations. A negative bias was found in forecast GHI for correctly forecast clear-sky cases and a positive bias in correctly forecast overcast cases. Temporal averaging improved the cloud cover forecast and hence decreased the solar radiation forecast error, but made little impact on the overall bias. The positive bias seen in overcast situations occurs when the model cloud has low values of liquid water path (LWP). We

attribute this bias to the model having LWP values that are too low or that the model optical properties for clouds with low LWP are incorrect.

## 1   Introduction

Accurate forecasts of solar radiation are valuable for solar energy, such as predicting the power generation one-day ahead for energy markets, and for public health reasons, such as forecasting the amount of UV radiation. The amount of solar radiation

at the surface is highly dependent on the solar zenith angle and clouds. However, clouds are highly variable in space and



time, as are their optical properties, therefore solar radiation forecasts require accurate cloud forecasts. Many applications only require reliable climatologies of the solar resource, such as solar resource assessments for solar energy installations (Kleissl, 2013). Observed climatologies can be obtained from surface-based instrumentation (Ohmura et al., 1998) and from satellite (Posselt et al., 2012; López and Batlles, 2014; Müller et al., 2015). Climatologies can also be derived from Numerical Weather

Prediction (NWP) forecasts and reanalyses, which are attractive from a cost perspective but may display larger uncertainties than observations (Jia et al., 2013; Boilley and Wald, 2015; Frank et al., 2018; Urraca et al., 2018). Climatologies require that the correct amount and type of cloud is predicted on average, whereas a forecast additionally requires that the cloud is forecast at the right time.

Evaluating cloud forecasts and their impact on solar radiation has been performed using ground-based observations; Ahlgrimm

and Forbes (2012) investigated the impact of low clouds on solar radiation in the Integrated Forecast System (IFS) by the European Centre for Medium-Range Weather Forecasts (ECMWF) at the Atmospheric Radiation Measurement (ARM) Southern Great Plains (SGP) site in the US using cloud radar, micropulse lidar and surface radiation measurements; Van Weverberg et al. (2018) investigated the positive temperature bias in the lower troposphere at SGP in nine different models, which was attributed to an overestimate of the net surface shortwave radiation arising from incorrectly modeled cloud radiative effects.

Earlier studies also suggest that supercooled liquid layers are not correctly represented in NWP models (Ahlgrimm and Forbes, 2012; Forbes and Ahlgrimm, 2014).

Continuous verification of the vertical representation of clouds in forecast models is available through Cloudnet (Illingworth et al., 2007), however, this requires comprehensive ground-based cloud observing systems, e.g ARM (Mather and Voyles, 2013) and Cloudnet, which are sparsely distributed across the globe. Verification of the column-integrated cloud amount (cloud

cover) can be performed at many more locations using operational SYNOP and/or ceilometer observations (Mittermaier, 2012). Ceilometers are much more widely distributed than cloud radars as they are also present at airports to detect clouds, especially liquid layers. Operationally most ceilometers only provide cloud base height and cloud amount, but, in principle, all ceilometers observe the attenuated backscatter profile. This profile can be further processed to yield information on the boundary layer, and the presence of aerosol, liquid, ice, and precipitation (Hogan et al., 2003; Morille et al., 2007; Münkel et al., 2007;

Van Tricht et al., 2014; Kotthaus and Grimmond, 2018). Manufacturer–provided cloud base algorithms are typically not public and have been developed for aviation purposes based on decreased visibility. Cloud base height has also been derived from a microphysical point-of-view from the attenuated backscatter profile (e.g. Illingworth et al., 2007; Martucci et al., 2010; Van Tricht et al., 2014). Our goal is to increase the cloud information available from the ceilometer attenuated backscatter profile and combine this with surface radiation measurements.

Ceilometers are often operated in large networks (e.g. by national weather services (Illingworth et al., 2015)) which are now being incorporated within harmonised pan-continental networks such as E-PROFILE (Illingworth et al., 2019), where the profile is being recorded. Thus, implementing ceilometer methods for evaluating cloud and radiation model forecasts would be a beneficial addition to the more comprehensive but sparse cloud profiling.

Our aim is to understand how the forecast of low and mid-level clouds in a NWP model impacts the forecast of solar radiation

at the surface. Moreover, our goal is a methodology that can be implemented rapidly at numerous sites with autonomous and





robust instrumentation, i.e. combining ceilometer and solar radiation observations (Section 2) with single-level fields from NWP models (single-level refers to surface fields and column-integrated fields). This requires accurate detection of liquid water clouds, precipitation, ice, and fog. In Section 3, we detail how we improved liquid cloud detection, and developed precipitation and fog identification algorithms, for ceilometers. In this study, we concentrated on evaluating the ECMWF IFS.

Details of the model, and the forecast cloud and solar radiation parameters investigated, are described in Section 4. Since we are comparing point measurements from the ceilometer and ground-based solar radiation instruments with the single-level output from gridded model data, both observations and forecast model parameters require post-processing before model evaluation. This post-processing methodology is presented in Section 5 and would be applicable to a wide range of NWP models and at hundreds of observation sites globally. We use four years of cloud cover and solar radiation observations from Helsinki,

Finland (Section 6) to investigate the skill of the IFS in forecasting clouds and radiation using our methodology (Sections 7–9), where we explicitly examine how the skill in forecasting cloud is related to the solar radiation forecast error.

## 2  Ceilometer and solar radiation observations

A ceilometer is an active instrument, which sends very short light pulses produced by a laser into the atmosphere and detects the backscattered signal from aerosol particles, cloud droplets and ice crystals. In this study we use a Vaisala CL51 ceilometer

for observing clouds, which has a wavelength close to 910 nm. Operationally, the instrument reports cloud base heights and cloudiness values (oktas), but the internal algorithms do not determine cloud type, such as whether the cloud contain liquid or ice, or both, and therefore, we do not use these the values. In addition to the standard cloud reporting, ceilometers can also provide the attenuated backscatter profile, from which it is possible to distinguish liquid layers, ice clouds, fog and precipitation; we describe the algorithms developed for this in Sect. 3. In this study, the vertical range resolution of the ceilometer is 10 m,

with attenuated backscatter profiles output every 15 seconds and a maximum range of 15 km. The calibration of the raw attenuated backscatter profiles is performed using the method of O'Connor et al. (2004), and the background noise is identified and removed based on the signal to noise ratio. The noise is calculated from the furthest range gates and assumed to be constant over the profile. The identification of high ice clouds is improved through temporal and spatial averaging to increase sensitivity, however, there are still challenges in identifying high ice clouds, especially during the day when the solar background noise is

high. Note that we take into account the ceilometer data post-processing methods recommended by (Kotthaus et al., 2016).

The ceilometer is suited to identification of liquid clouds and precipitation in the vertical profile, however, the measurement is usually limited to the lowest liquid cloud layer due to strong attenuation, and no information is available above this layer. Figure 1a shows an example of calibrated, background-noise-removed ceilometer attenuated backscatter profiles during 9 hours at Helsinki, Finland on 30 March 2016. A fog layer has been identified from 08:00 UTC to 09:45 UTC with no information

available above. Liquid cloud layers have been identified between 10:30–11:00 UTC (below 1 km) and 11:00–13:30 UTC (below 2.5 km), again with no information available above, except around 12:30 when the liquid layer is dissipating. The signal is also attenuated in the case of heavy precipitation in which the ceilometer may not detect the cloud base above the precipitation layer. Precipitation, here in the form of ice, is clearly visible in Fig. 1a 10:00–10:30 UTC, 13:30–16:00 and after



16:30 UTC, and does not reach the ground. Weak backscatter from aerosol in the boundary layer (orange color) is visible when there is no precipitation, fog, or liquid layers close to the ground. Since the ceilometer reliably detects the first cloud layer, we can use the data to derive robust cloud cover quantities even though we cannot say if there is any more cloud above the first layer detected.

5      Solar radiation, specifically Global Horizontal Irradiance (GHI), is measured with a Kipp & Zonen CM11 Secondary Standard pyranometer. Automated quality control has been applied by the Finnish Meteorological Institute (FMI) together with a visual check to ensure the data quality. The automated quality control is based on the Baseline Surface Radiation Network (BSRN) quality control procedure (Long and Shi, 2008) with small modifications to be more suitable for Finnish conditions (Rontu and Lindfors, 2018). GHI measurements are stored as one-minute averages in the FMI database.

## 3   Ceilometer algorithm development

### 3.1   Liquid layer identification improvements

In this study, we develop an algorithm to detect liquid cloud layers. The Cloudnet (Illingworth et al., 2007) approach for detecting the liquid cloud base is used as a starting point. The Cloudnet approach relies on the shape of the attenuated backscatter profiles, as it is known that the liquid droplets result in high backscatter signal and the signal attenuates in the liquid layer (Fig. 1c). Thus, liquid layers display local peaks of stronger signal in the vertical profile of attenuated backscatter coefficient $\beta$. The Cloudnet approach searches for the lowest height range gate where the attenuated backscatter value exceeds the given threshold ($\beta = 2 \cdot 10^{-5}$ m$^{-1}$ sr$^{-1}$, representing liquid and called as a pivot) and where the signal is attenuated 250 m above the pivot value. If the signal attenuates above the pivot value, the cloud base is found below the pivot value based on the gradient in the $\beta$ profile. Multiple liquid cloud bases are allowed in the Cloudnet method. This method is part of the Cloudnet approach for identifying "droplet bits" within the categorization process (Illingworth et al., 2007) and is described in detail here: http://www.met.rdg.ac.uk/ swrhgnrj/publications/categorization.pdf, under the Section "3.4.2 Droplet bit".

The Cloudnet approach is skillful in situations when there is no precipitation. During strong precipitation the attenuated backscatter coefficient may exceed the given threshold used in the Cloudnet droplet bit algorithm, even if stronger values representing the true liquid layer would be present above. Therefore, the cloud base may incorrectly be identified inside the precipitation layer below the true liquid cloud base (Fig. 2a). The liquid cloud base might not be always visible due to attenuation of the signal in heavy precipitation layer. We improved the method to enable reliable detection in all cases, including heavy precipitation.

The algorithm for finding liquid layers relies on the same principles as the Cloudnet approach. However, our approach for finding the strong $\beta$ value (pivot), representing the liquid layer, differs. Our updated liquid layer identification relies more on the shape of the profile than an absolute threshold value, and the fact that a liquid layer exhibits a strong peak in the attenuated backscatter profile. Therefore, the maximum of a localized peak value of $\beta$ is found (not only the first value above a certain threshold), with the requirement that the magnitude of the local maximum exceeds the same threshold $\beta$ value as in the Cloudnet approach. An additional requirement is that the peak width is not too broad with the maximum peak width at

(c) Author(s) 2018. CC BY 4.0 License.





half-height being set to 150 meters. This ensures that the identified peak is attenuating rapidly (O'Connor et al., 2004) rather than the relatively weak attenuation expected in precipitation so that threshold exceedance found in precipitation is not enough to trigger false liquid layer identification. The cloud base below the strong $\beta$ value is found using the same method as for the Cloudnet droplet bit algorithm.

Visual validation of our updated algorithm is shown in Fig. 2, which confirms that liquid cloud layer identification during precipitation is more accurate. The Cloudnet processing suite will soon be updated with this new algorithm, which will also improve Cloudnet-derived products. This new algorithm can be used for other applications such as the identification of liquid layers for in-cloud icing detection for wind turbine operators and aviation.

## 3.2   Precipitation and fog identification

In addition to liquid layers, we require fog, precipitation, and ice cloud identification. The profiles in these conditions show particular characteristics (Fig. 1b–d). Precipitation, including ice (we assume that all ice is falling), is identified from the shape of the attenuated backscatter profile (Fig. 1d). We identify the base of the precipitating layer, which, in practice, means the altitude where the precipitation is either evaporating or reaching the ground. Typically, attenuated backscatter coefficient values are lower for precipitating rain and ice, relative to liquid droplets. This is due to their much lower number concentrations even

though the particle sizes are larger. The ceilometer signal is not attenuated as rapidly during precipitation and the ceilometer can "see" further into the precipitation. The precipitation algorithm uses a threshold value of $\beta = 3 \cdot 10^{-6}$ m$^{-1}$ sr$^{-1}$, determined to be suitable in this study, together with a layer thickness greater than 350 m (i.e. the ceilometer backscatter signal is not attenuated within 350 m). We determined these thresholds by visual analysis. The layer base is simply the lowest range gate where these two conditions are satisfied. Both precipitation and a liquid layer can be identified within the same profile.

Fog at the surface cannot always be identified using the liquid layer identification method, which relies on finding a local maximum in the $\beta$ profile. An example of fog is given in Fig. 1b where there are already high $\beta$ values in the first range gate. Here we check the rate of the attenuation above the fog layer maximum as it may not be possible to define a peak. The threshold for fog is set as $\beta = 10^{-5}$ m$^{-1}$ sr$^{-1}$, with a $\beta$ value 250 m above the instrument of $\beta < 3 \cdot 10^{-7}$ m$^{-1}$ sr$^{-1}$.

## 4   Model data

### 25   4.1   The Integrated Forecast System (IFS)

Forecasts produced by the Integrated Forecast System (IFS), run operationally by the European Centre for Medium-Range Weather Forecasts (ECMWF), are analysed in this study. The IFS is a global numerical weather prediction (NWP) system which includes observation processing and data assimilation in addition to the forecast system. The IFS is used to produce a range of different forecasts, from medium range to seasonal predictions, and both deterministic and ensemble forecasts.

In this study we only consider the high resolution deterministic medium-range forecasts (referred to as HRES), which have a horizontal resolution of approximately 9 km and 137 vertical levels. The vertical grid spacing is non-uniform and below



15 km varies from 20 to 300 m with higher resolution closer to the ground. The temporal resolution of the model output is one hour and forecasts up to 10 days in length are run every 12 hours. A full description of the IFS can be found from ECMWF documentation: https://www.ecmwf.int/en/forecasts/documentation-and-support/changes-ecmwf-model/ifs-documentation.

The IFS is under constant development and typically a new version becomes operational every 6–12 months. Therefore,
unlike reanalysis, which is based on a static model system, the archived forecasts from the operational IFS reflect changes in the model. Although the aim of this paper is not to quantify how changes to the IFS affect the cloud and solar radiation forecasts, a brief overview of model updates is given here.

Several upgrades have been implemented into the IFS during the four year (2014–2017) data period that is used in this study (all are described in the IFS documentation: https://www.ecmwf.int/en/forecasts/documentation-and-support/changes-
ecmwf-model/ifs-documentation.) A major upgrade occurred in March 2016 when the horizontal grid was changed from a cubic-reduced Gaussian grid to an octahedral-reduced Gaussian grid, resulting in an increase in horizontal resolution from 16 km to 9 km. The cloud, convection, and radiation parameterizations schemes strongly influence the forecast of clouds and radiation and all of these schemes have undergone updates during the four year period considered here. Notably, the radiation scheme was updated from McRad scheme (Morcrette et al., 2008) to the computational cheaper ECRAD scheme (Hogan and
Bozzo, 2016) in 2016 meaning that the radiation scheme is now called more frequently. Aerosols also impact radiation forecasts and are represented in the IFS by a seasonally varying climatology. In July 2017 the aerosol climatology was updated to one derived from the aerosol model developed by the Copernicus Atmospheric Monitoring Service and coupled to the IFS (Bozzo et al., 2017). Note that in the current version of the IFS aerosol and clouds do not interact.

## 4.2   Model output used in this study

We use day-ahead forecasts, which have been initialised at 12:00 UTC the previous day and corresponding to forecast hours t+12 to t+35, obtained from the closest land grid point to the measurement site. Day-ahead forecasts are commonly used in the solar energy field for estimating the daily production for the energy market. A list of the model variables we use is given in Table 1.

One goal is to develop simple and robust methods for evaluating the skill that the model has in forecasting clouds and solar
radiation, which can be rapidly applied to numerous sites globally. Therefore, we take the single-level cloud forecast variables: low cloud cover (LCC; Table 1) and medium cloud cover (MCC). These are defined in the IFS as follows: low is model levels with a pressure greater than 0.8 times the surface pressure (from ground to approximately 2 km in altitude); medium encompasses model levels with a pressure between 0.45 and 0.8 times surface pressure (approximately 2–6 km). For IFS, the cloud layer overlap is also taken into account when calculating LCC and MCC, the degree of randomness in cloud overlap is
a function of the separation distance between layers (the greater the distance between layers, the more randomly overlapped they are, Hogan and Illingworth, 2000). For solar radiation forecasts, we use the surface solar radiation downwards (SSRD), which is a single-level parameter output hourly as an accumulated value (from the start of the forecast) in units of $J\,m^{-2}$.

Other model variables are also downloaded for further calculation and for more detailed investigation of the sources of forecast error. Pressure (PRES) on model levels and surface pressure (SP) are used to determine the altitude levels for low and



medium cloud cover classes for ceilometer data post-processing. Temperature (T) on model levels is used for classifying warm and cold (supercooled) liquid clouds. Specific cloud liquid water content on model levels (CLWC), provided as a mixing ratio, is used to calculate the total cloud liquid water path (LWP).

## 5   Methods for evaluating the model performance

Some further calculation is needed in order to evaluate the model output against the observations, as the variables obtained from the model and observations are not directly comparable. The forecast cloud cover is a single-level variable representing instantaneous values of column-integrated cloud coverage over an area (model grid of approximately 16x16 km before the resolution upgrade and 9x9 km area after the resolution upgrade) with hourly resolution. The ceilometer attenuated backscatter profile observations are point measurements with high temporal resolution (15 s), from which cloud occurrence can be derived.

The forecast solar radiation is an accumulated value in $\mathrm{J\,m^{-2}}$ since the beginning of the forecast, whereas the observed GHI (in $\mathrm{W\,m^{-2}}$) is a point measurement averaged to one-minute resolution. Post-processing of both forecast and observations is required to obtain a comparable dataset, discussed in the following subsections. After further post-processing, skill scores are then used to evaluate the cloud cover forecasts, and different error metrics are used to calculate the solar radiation forecast error.

In this study, we only consider daytime hours for model evaluation as our focus is on solar radiation forecasts. Therefore, hours with hourly-averaged GHI measurements less than 5 $\mathrm{W\,m^{-2}}$ are removed. For northern latitudes, this results in a range from 2 to 19 hours per day, depending on the season (short days in winter and long days in summer). Furthermore, it is required that the data availability of observations over each hour is at least 75 %; otherwise the hour is discarded from the analysis.

### 5.1   Post-processing of cloud cover forecasts and ceilometer observations

The difference arising from the fundamental differences of cloud information obtained from model (grid value) and observations (point measurement) must be compensated. As the clouds are advected over the measurement site, the temporal average of the point measurements of cloud occurrence is correlated to the cloud cover over an area. Therefore, averaging the ceilometer observations over certain time window is assumed to correspond to cloud cover represented in grid space. The suitable averaging time window for cloud cover may not be easy to define; here one-hour averages are used as this is the temporal resolution of the model output. The horizontal resolution of the model is 16 km/9 km, and therefore one-hour averaging corresponds to advection speeds of 4.5 $\mathrm{m\,s^{-1}}$ or 2.5 $\mathrm{m\,s^{-1}}$. However, we are aware that this averaging procedure may not always be appropriate for comparison and is kept in mind when analysing the results.

High and thin ice clouds are not reliably detected with ceilometers (see Section 2), therefore we only consider clouds at low to medium altitudes in both the model and observations. We do not evaluate the model total cloud cover (TCC), as this contains

contributions from high clouds.

The model variables LCC and MCC account for cloud within their relevant height ranges regardless of whether there is cloud in a lower level. In contrast, the ceilometer usually only detects the base of the first cloud layer. For example, the ceilometer





may detect a cloud base to be below 2 km, hence defining it as low cloud, but the cloud may also contribute significantly to mid-level cloud cover, which is not captured by the ceilometer. Additionally, a cloud base may not be detected in strong precipitation due to the attenuation of the lidar signal. In these cases, the bottom of the precipitation layer is treated as a cloud base, even though in reality the cloud producing the rain is at higher altitude. Thus, we combine low and medium cloud cover,
rather than investigating them separately.

Cloud cover is estimated from the ceilometer data as follows: first, the attenuated backscatter profiles are averaged over one minute before applying the algorithms described in Section 3. Then, liquid layers, precipitation (including ice clouds) and fog are identified for each one-minute profile. The forecast pressure on model levels is interpolated to the ceilometer range gate heights using the model height (ECMWF uses a terrain-following eta-coordinate system). Cloud cover at each level (low and
medium, defined in terms of pressure as for the model) is calculated as the percentage of cloud occurrence (occurrence of liquid cloud, precipitation/ice cloud, or fog) within each level over each hour. Finally, the observed cloud cover is the hourly sum of the observed low and medium cloud cover. Note that here, the observed cloud cover is a summation since it is calculated from time series of independent columns where only the first cloud layer contributes to the cloud cover calculation (the lowest layer).

The forecast LCC and MCC represent the fractional cloud cover (from 0 to 1) over the grid point and combining these requires an assumed overlap factor. In this study we use the random overlap assumption, which may result in a slight overestimate.

## 5.2 Post-processing of solar radiation forecasts and solar radiation observations

Forecast surface solar radiation (SSRD) is compared against the observed Global Horizontal Irradiance (GHI). Values of SSRD
require de-accumulating to hourly averages as the forecast solar radiation is an accumulated field from the beginning of the forecast and are transformed from $J\,m^{-2}$ to $W\,m^{-2}$. Observed 1-minute averaged GHI measurements ($W\,m^{-2}$) are averaged over one hour for comparison.

## 5.3 Skill scores for cloud cover forecasts and error metrics for solar radiation forecast error

Cloud cover forecasts are evaluated with 2D-histograms and skill scores. We use the Mean Absolute Error Skill Score (MAESS;
Hogan et al. (2009)) and Mean Squared Error Skill Score (MSESS; Murphy (1988)), which compare the occurrence of a cloud separately in observations and in forecasts, and take into account the magnitude of the difference. MAESS uses the absolute difference between forecast and observed value, and MSESS uses the squared difference, therefore penalizing larger errors more than small but more common differences. The skill scores are based on the contingency table (Table A1 in Appendix A1), where the occurrence of hits, false alarms, misses and correct negative values by given cloud cover threshold are calculated. For
example, a hit occurs when both forecast and observed cloud cover are above a given cloud cover threshold. Here, the threshold for a cloud cover is set to 0.05, following the method used by Hogan et al. (2009). Therefore, a hit means that some amount of cloud is both forecast and observed, however, a hit does not yet imply a perfect forecast. For both MAESS and MSESS, the skill of a random forecast is 0 and a perfect forecast, 1. The equation for MAESS and MSESS is given in Appendix A1.





The error metrics Mean Absolute Error (MAE), Mean Absolute Percentage error (MAPE), Mean Biased Error (MBE), and Root-Mean-Square-Error (RMSE) are used to evaluate the solar radiation forecast errors. These error metrics are defined in Appendix A2. MAE, RMSE and MBE are absolute error metrics and result in forecast error in $\mathrm{W\ m^{-2}}$, whereas MAPE is a relative error given in %. MBE is the only error metric that shows the sign of the error. A positive bias is seen when the model overestimates the incoming surface solar radiation, whereas a negative bias is when the model underestimates the incoming solar radiation.

## 6  Site characteristics and cloud and radiation climatology

The measurement site is located on the roof of FMI in Helsinki, Finland (60.204° N, 24.961° E, Fig. 3, measurements at 26 m above sea level), located less than 10 km from the coastline of the Gulf of Finland. Coastal effects, such as sea breezes, are common. There are no large variations in topography around the site.

We investigate the cloudiness and solar resource at this site using four years of ceilometer observations and solar radiation measurements. There is an annual variation in the observed cloudiness at the site (Fig. 4a) with overcast conditions (cloud cover $\geq 0.95$) being more common in winter and less common in summer. In contrast, broken cloud ($0.05 <$ cloud cover $<$ 0.95) and clear (cloud cover $\leq 0.05$) conditions are most common in summer and least common in winter. The variation in cloudiness is quite high from year to year, especially in summer, but, in winter the most probable sky condition contains cloud.

In addition to the observed annual variation of cloudiness, the observed annual variation of incoming solar radiation is strongly influenced by the northern location of the site (60° N). The length of the shortest day of the year (winter solstice on 21st or 22nd December) is less than 6 hours and the length of the longest day (summer solstice between 20th and 22nd June) is almost 19 hours. Thus, the amount of solar radiation at the top of the atmosphere is much higher during summer (Fig. 4b, solid line). This signal is also clear in the amount of solar radiation reaching the ground, the measured GHI (Fig. 4b, dashed line), which is dependent on both the incoming solar radiation at the top of atmosphere and the attenuation of the downward flux due to clouds and the atmosphere. The year-to-year variation in the monthly mean of measured GHI is much greater during summer months (lighter shaded area in Fig. 4b), with variations reaching $140\ \mathrm{W\ m^{-2}}$ in August, which is larger than the monthly mean GHI during winter months.

To investigate the seasonal variation, we define seasons based on the annual variation in the solar resource (Fig. 4b). The summer season is defined as May to July when the solar resource is at a maximum, and winter is defined as November to January, when the solar resource is at a minimum (Spring is February to April, Autumn is August to October).





## 7 Forecast skill in predicting clouds and radiation

### 7.1 How well are clouds forecast?

We now investigate how well the IFS forecasts clouds over our site in Helsinki, Finland. Since we are interested in the solar resource we only evaluate time steps where the hourly-averaged observed GHI is greater than 5 W m$^{-2}$ to link the skill in

forecasting clouds to the skill in forecasting radiation. (Sect. 7.3).

In Fig. 5 we compare the observed and forecast cloud cover for each season. For a perfect forecasts, all values would lie on the diagonal (dashed line) in each scatter plot. For all seasons, the majority of cloud cover values are concentrated around clear conditions (pair 0;0) and overcast conditions (pair 1;1) for both observations and forecasts. This suggests that not only are clear and overcast conditions the most commonly observed, but also most skillfully forecast in all seasons. During winter

the vast majority of cloud cover observations and forecasts are at (or close to) being overcast (Fig. 5a). Clear sky conditions are more common in other seasons (both observations and model). The large spread for both observed and forecast cloud cover values between 0.1 and 0.9 indicates that partly cloudy conditions are challenging for the IFS to correctly predict. However, these cases are not as common as clear and overcast cases, which is a result of observed and forecast cloud cover distributions being strongly U–shaped for typical NWP model grid-sizes (Hogan et al., 2009; Mittermaier, 2012; Morcrette et al., 2014).

It is also notable that, during all seasons, there are values on the boundaries of the scatter plot away from the diagonal, for example, where the model is incorrectly forecasting clear sky during cloudy conditions, or overcast conditions during clear or broken skies. Summer and Autumn seasons (Figs. 5c,d) display more broken cloud conditions, also seen in Fig. 4a, when the solar resource is high (Fig. 4b).

Skill scores represent the model's ability to forecast a given variable. To calculate skill scores, we generate a contingency

table for cloud cover. This requires a binary forecast so we use a threshold cloud cover value of 0.05 as in Hogan et al. (2009) to define the presence of cloud: a hit is cloud observed and forecast; false alarm, cloud not observed but forecast; miss, cloud observed but not forecast; correct negative, cloud not observed nor forecast.

The annual relative occurrences of contingency table elements (hit, false alarm, miss, correct negative) are shown in Fig. 6a. During all months, hit has the highest relative occurrence (mean 68 %), indicating that the model usually contains some low

or mid level cloud when cloud is also observed at these levels. The hit occurrence is greatest between October and February, when overcast conditions are also most common (Fig. 4a). Note that a hit requires that both observations and model has some cloud, but it does not necessarily represent a perfect forecast. Similarly, the relative occurrence of correct negative is highest during spring and summer months. False alarms are most common in summer and autumn when their relative occurrence reach 17 %. The relative occurrence of missed clouds is low (mean 4 %) for all months and there is no clear seasonal cycle.

Skill scores are then generated from the contingency table; we use MAESS and MSESS as these take into account the magnitude of the difference between the observed and forecast cloud cover (Fig. 6b,c). MAESS and MSESS both show annual variation, being highest during winter months and lowest during summer months. This information is important, especially for solar energy purposes, as it shows that clouds are forecast less skillfully in summer, which is when the solar resource is





greatest. There are also notable variations in skill scores from year to year, especially in October and December. MSESS is greater than MAESS, especially during summer when more broken cloud conditions are expected.

## 7.2 How well is solar radiation forecast?

As expected, there is a large seasonal variation in observed GHI, up to $900 \, \text{W m}^{-2}$ in summer (Fig. 7c) and less than $300 \, \text{W m}^{-2}$ in winter (Fig. 7a). The absolute error in the solar radiation forecast can potentially therefore be much higher in summer, and is evident in the potential range of scatter between observed GHI and forecast GHI for each season (Fig. 7). The forecast of solar radiation is usually overestimated in all seasons (Fig. 8), especially for low irradiance values where the positive bias is more obvious. Solar radiation forecast MAE (Fig. 8a, solid line) is greater in summer than in winter, as is the year-to-year variation in monthly absolute errors (shaded area in Fig. 8a). There is no clear seasonal cycle in the variation in the relative error (MAPE) from year to year, however, MAPE itself peaks in February and November.

The mean biased error (MBE) in the solar radiation forecast is positive when the model overestimates solar radiation at the surface. Figure 8b shows separate calculations of the monthly mean positive (red) and negative (blue) bias in forecast GHI. Throughout the year, the positive bias (both absolute and relative) is greater than the negative bias, thus the model overestimates solar radiation more than it underestimates. The year-to-year variation in relative positive MBE is also larger than the relative negative MBE. For example, the relative positive MBE in solar radiation forecast ranges between 50 % and 125 % whereas the relative negative MBE is rather constant at around 25 %. Overestimates are also more common than underestimates (not shown). The result is an overall positive bias in forecast GHI. Negative relative MBE is constant throughout the year, both positive MBE metrics show the same seasonal response as the corresponding MAE/MAPE metric and negative MBE shows the same summer enhancement as positive MBE but with the opposite sign.

## 7.3 How do errors in cloud cover impact the solar radiation forecast?

Assuming the correct representation of radiative transfer in the atmosphere, with only the forecast of cloud impacting the solar radiation forecast at the surface, an increase in forecast cloud cover would be expected to result in a reduction in the amount of forecast solar radiation. However, the amount of cloud may be correctly forecast, but not the cloud properties. Since cloud properties are directly responsible for cloud radiative properties, both cloud amount and properties should be correctly forecast in order to obtain a reliable solar radiation forecast.

Figure 9 shows the annual cycle of accumulated positive and negative bias in the cloud cover forecast and solar radiation forecast. It can be seen that months with a large accumulated negative bias in cloud cover forecasts (e.g. June 2014) show a notably large accumulated positive bias in the solar radiation forecast. However, not all months show a clear correlation between a negative bias in the cloud cover forecast and a positive bias in the solar radiation forecast. This is most probably due to compensating effects where, for example, the cloud cover forecast could be overestimated (positive bias in cloud cover) but the liquid water content forecast is underestimated (would result in positive bias in solar radiation forecasts).

To investigate how well the forecast cloud cover corresponds to the observed cloud cover the counts of hourly observed and forecast cloud cover values are paired together in 2D-histograms (Fig. 10a). For perfect forecasts, all counts would lie on the





diagonal. Figure 10a shows that there are many correctly forecast situations for clear sky (0;0) and overcast (1;1). However, it is clear that there are many values on the boundaries, which means that cloud is either observed and not forecast (miss), or cloud is forecast but not observed (false alarm). At one hour resolution, 47 % of the total number of counts are above the diagonal, thus the forecast cloud cover is overestimated on average. The forecast underestimates cloud cover 34 % of the time. Note that

changing the overlap assumption from random to maximum when calculating the combined cloud cover (LCC+MCC) changes these values by 3 %.

The solar radiation forecast MBE for concurrent pairs of cloud cover values in Fig. 10a is presented in Fig. 10b. MBE values below the diagonal, where the forecast cloud cover is underestimated, are mostly positive; similarly MBE values above the diagonal are mostly negative, where the forecast cloud cover is overestimated. Note that the change from positive to negative

MBE does not quite follow the diagonal, with minimal bias appearing to follow a line from (0.1;0) to (0.8;1), i.e. observed cloud cover greater than 0.9 shows a positive solar radiation forecast MBE (27 W m$^{-2}$) and observed cloud cover less than 0.1 shows negative MBE (-16 W m$^{-2}$). This negative bias during clear sky situations over Helsinki was also observed by Rontu and Lindfors (2018), and is most likely due to the aerosol climatology implemented in the model having too much aerosol . Another possible source of negative bias during clear sky situations would be too much water vapor in the atmosphere. There

are earlier studies showing similar results elsewhere (Ahlgrimm and Forbes, 2012; Frank et al., 2018). Overcast situations occur more frequently (23 % of the time), resulting in the overall positive bias in the solar radiation forecast.

## 8   Impact of temporal averaging

Forecasting individual clouds in the right place at the right time is challenging and here we investigate whether temporal averaging improves the cloud forecast and therefore the radiation forecast. Different averaging windows (3-hourly, 6-hourly,

12-hourly, daily) are used in preparing the data for evaluation in the same manner as for Fig. 10a and the results for selected averaging-windows are shown in Fig. 11a–c. The agreement between observed and forecast cloud cover improves with increasing averaging windows, and the number of cases of extreme misses and false alarms (corners (1;0) and (0;1)) reduces.

When calculated separately, the magnitudes of the positive and negative solar radiation forecast MBE for concurrent pairs of cloud cover values decrease with increasing averaging time. The mean positive bias decreases from 65 W m$^{-2}$ when averaging

over one hour to 35 W m$^{-2}$ when averaging over one day, the mean negative bias reduces from -46 W m$^{-2}$ to -27 W m$^{-2}$. Temporal averaging has little impact on the overall bias (8 W m$^{-2}$ for hourly average, 9 W m$^{-2}$ for daily average). Increasing the averaging window does not alter the pattern where the change from positive to negative MBE is away from the diagonal. The negative bias in clear sky conditions and positive bias in overcast conditions are still present, suggesting that the bias is likely to be due to cloud properties rather than the cloud presence.

Figure 12 summarizes the impact of temporal averaging on the skill in forecasting cloud cover and the error in forecasting solar radiation, with skill clearly increasing, and error reducing, as the averaging window is lengthened. Extreme misses and false alarms for cloud cover are reduced, and for GHI MAE, the individual absolute errors are reduced with temporal averaging. Persistence forecasts were also investigated; a persistence forecast uses the forecast for the day before. The skill for the cloud



cover persistence forecast also increases with increasing temporal averaging, as does the reduction of error in the persistence GHI forecast, however, these are not as good as the actual forecasts at this location.

## 9   Overcast analysis

Figures 10 and 11 show a positive bias in the solar radiation forecast even when overcast conditions are correctly forecast, for

all averaging windows. As the cloud amount is correctly forecast, this suggests that the bias must be due to cloud properties. We investigate the forecast cloud base temperature and cloud liquid water path (LWP). Previous studies have shown that clouds containing supercooled liquid (T < 0 $°C$) are poorly forecast (Forbes and Ahlgrimm, 2014; Barrett et al., 2017), and LWP is one parameter that contains information on the amount of liquid water in a cloud, directly impacting how much solar radiation is transmitted through the cloud.

We consider correctly forecast overcast cases (observed and forecast cloud cover > 0.9) containing liquid. The clouds are classified as warm or cold (supercooled), depending on their cloud base temperature, using the temperature profile from the IFS as no observed temperature profiles are available. We then bin the clouds based on their forecast cloud LWP obtained by integrating the forecast cloud liquid water content (clwc, Table 1). We selected three bins representing relatively high (LWP > 0.2 kg m$^{-2}$), moderate (0.2 kg m$^{-2}$ ≥ LWP ≥ 0.05 kg m$^{-2}$) and low (LWP < 0.05 kg m$^{-2}$) cloud liquid water content. These

values were selected based on the range of optical depths that would be expected for each LWP range bin. Unfortunately, there was no observed LWP available for this measurement site.

Figure 13 shows that the positive bias in the solar radiation forecast increases with decreasing LWP. Note that the response is similar for both warm and cold liquid clouds. For warm clouds, the MBE in GHI increases from 16 W m$^{-2}$ for clouds with high LWP to 70 W m$^{-2}$ for clouds with low LWP. For cold clouds, the MBE in GHI increases from 15 W m$^{-2}$ for clouds with

high LWP to 36 W m$^{-2}$ for clouds with low LWP. This suggests that either forecast clouds do not have enough LWP or that the optical properties of clouds with low LWP are not properly modeled. The first conclusion, that forecast clouds do not have enough LWP, is consistent with the findings of Ahlgrimm and Forbes (2012). They found a positive bias in ECMWF IFS for overcast situations with low cloud at the Atmospheric Radiation Measurement site in the Southern Great Plains. Furthermore, they found that IFS overestimates the occurrence of clouds with low LWP and underestimates the number of clouds with high

LWP, which also results in a positive bias in solar radiation forecasts. Challenges in correctly modeling supercooled liquid clouds have previously been reported, but our results suggest that the issue of a positive bias in GHI is more pronounced for warm clouds, and not just an issue for supercooled liquid clouds.

Also of interest is that the relative bias in GHI is constant across a wide range of GHI values. This implies that a simple LWP-dependent correction factor could be applied to the GHI forecast to remove the observed bias.





## 10 Conclusions

We have used ceilometer and solar radiation measurements to evaluate the cloud cover and solar radiation forecasts in the ECMWF operational IFS model over Helsinki, Finland. To obtain reliable cloud cover information from the ceilometer attenuated backscatter profiles, we took the Cloudnet liquid bit algorithm (Illingworth et al., 2007) as a starting point, updated the liquid cloud detection, especially during precipitation events, and developed additional algorithms for discriminating fog, precipitation and ice. The new algorithms are widely applicable for both operational use and research, e.g. in-cloud icing detection for the wind energy industry and for aviation. The updated algorithm will also be implemented operationally throughout the ACTRIS–Cloudnet network.

Over Helsinki, both observed and forecast cloud cover distributions are U–shaped indicating that most of the time the sky is either clear or overcast. Overcast conditions are most common in winter, whereas clear (and broken cloud) conditions are more common in summer. Cloud cover is better forecast in winter, however, this is when the solar resource is lower. The measured GHI is strongly influenced by the annual solar resource characterized by the northern latitude and annual variations in cloudiness; the absolute solar radiation forecast error tracks GHI, however the relative error is more or less constant throughout the year.

As expected, the bias in forecast GHI is negative when the model overestimates cloud cover (incoming solar radiation is underestimated by the model) and positive when the model underestimates cloud cover. Temporal averaging of the data improves the cloud cover forecasts and decreases the solar radiation forecast errors, as was shown by Hogan et al. (2009). The mean overall bias in the GHI forecast is positive ($8 \, \text{W} \, \text{m}^{-2}$). However, there is a negative bias in forecast GHI for correctly forecast clear cases and a positive bias in correctly forecast overcast cases. A mean overall positive bias would be expected if, on average, the forecast cloud cover was being underestimated, but, the forecast cloud cover is usually overestimated on average. This is because the positive GHI bias for the very frequent overcast situations dominates the overall bias. This positive bias occurs for cases where the model cloud has low values of LWP, and we attribute this bias to the model having LWP values that are too low or that the model optical properties for clouds with low LWP are incorrect.

In the future, these methods and analysis can be extended to hundreds of sites across Europe which are now producing ceilometer attenuated backscatter profiles. This analysis will also be performed at Cloudnet stations, which have the advantage in that they have observations of LWP, together with full cloud profiling, enabling the source of positive bias in clouds with low LWP to be investigated further.



## Appendix A: Skill scores and error metrics

### A1 Skill score calculation

**Table A1.** Contingency table for skill score calculation. Total number of counts, $n = a + b + c + d$, where $a$, $b$, $c$, and $d$ are the number of counts for each situation.

|  | Observed cloud cover $> 0.05$ | Observed cloud cover $\leq 0.05$ |
| --- | --- | --- |
| Forecast cloud cover $> 0.05$ | a = Hit | b = False alarm |
| Forecast cloud cover $\leq 0.05$ | c = Miss | d = Correct negative |

The skill scores in this study are calculated using the generalized skill score equation (Hogan et al., 2009), which, for MAESS and MSESS can be simplified to

$$S = 1 - \frac{x}{x_r},\qquad\text{(A1)}$$

where $x = (frc - obs)^2$ for MSESS and $x = |frc - obs|$ for MAESS, and the values for the random forecast, $x_r$, are calculated from elements of the contingency table: $x_r = \frac{a+b}{n} \cdot \frac{a+c}{n} + \frac{d+c}{n} \cdot \frac{d+b}{n}$ for both MSESS and MAESS. The values $obs$ and $frc$ refer to the observed and forecast values of the variable of interest, e.g. cloud cover.

### A2 Error metrics

$$MAE = \frac{1}{n}\sum_{t=1}^{n}|frc - obs| \qquad\text{(A2)}$$

$$MAPE = \frac{1}{n}\sum_{t=1}^{n}\left|\frac{frc - obs}{obs}\right| * 100 \qquad\text{(A3)}$$

$$MBE = \frac{1}{n}\sum_{t=1}^{n}(frc - obs) \qquad\text{(A4)}$$

$$RMSE = \sqrt{\frac{1}{n}\sum_{t=1}^{n}(frc - obs)^2} \qquad\text{(A5)}$$

*Acknowledgements.* This work was supported by the Maj and Tor Nessling Foundation (grant 201700032) and by the European Commission via project ACTRIS2 (grant agreement no. 654109). VAS is funded by the Academy of Finland (project no. 307331). We acknowledge ECMWF for providing model output from IFS, and FMI for providing ceilometer and solar radiation observations.



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



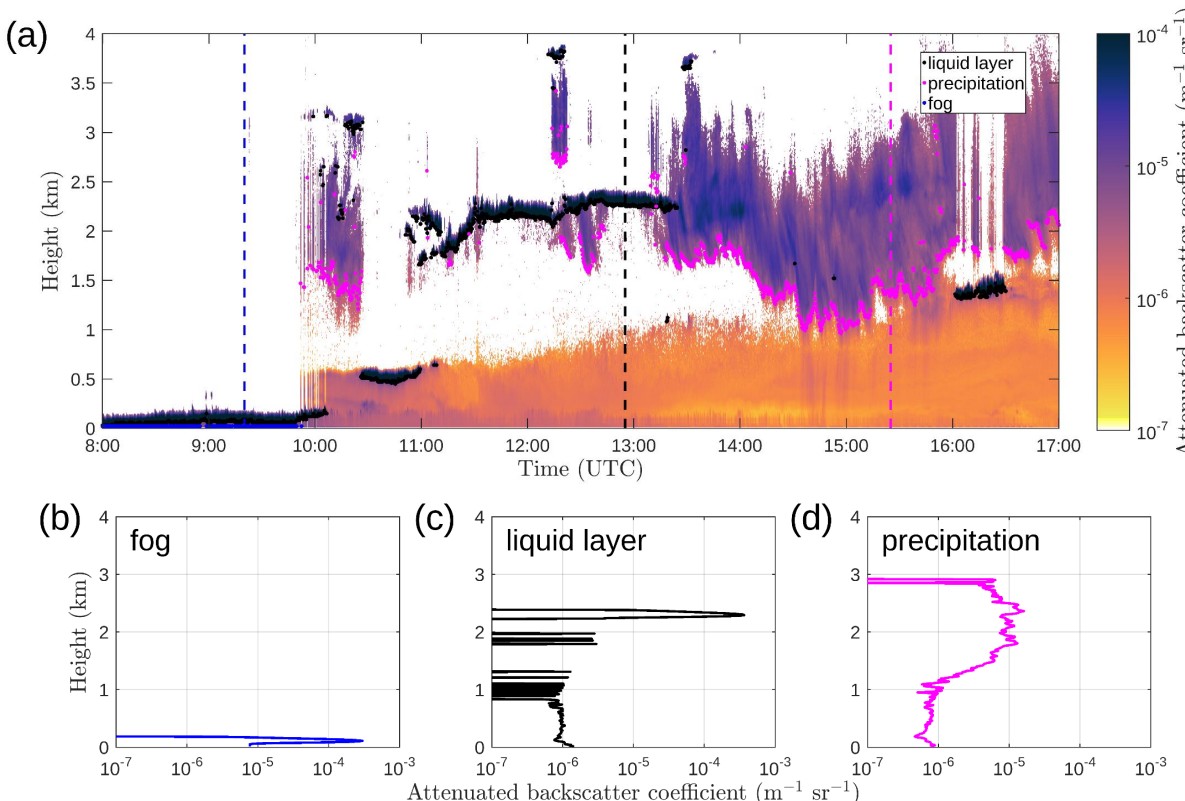

**Figure 1.** Time-height cross section of attenuated backscatter profiles from a Vaisala CL51 ceilometer on 30th March 2016 at Helsinki, Finland (a). Overplotted are the results from our identification algorithms: fog (blue dots), liquid cloud base (black dots), and precipitation base (magenta dots). Sample attenuated backscatter profiles are also shown for fog (b), liquid cloud layer (c), and precipitation (d). Dashed lines in (a) show the time when the profiles (b–d) are measured.



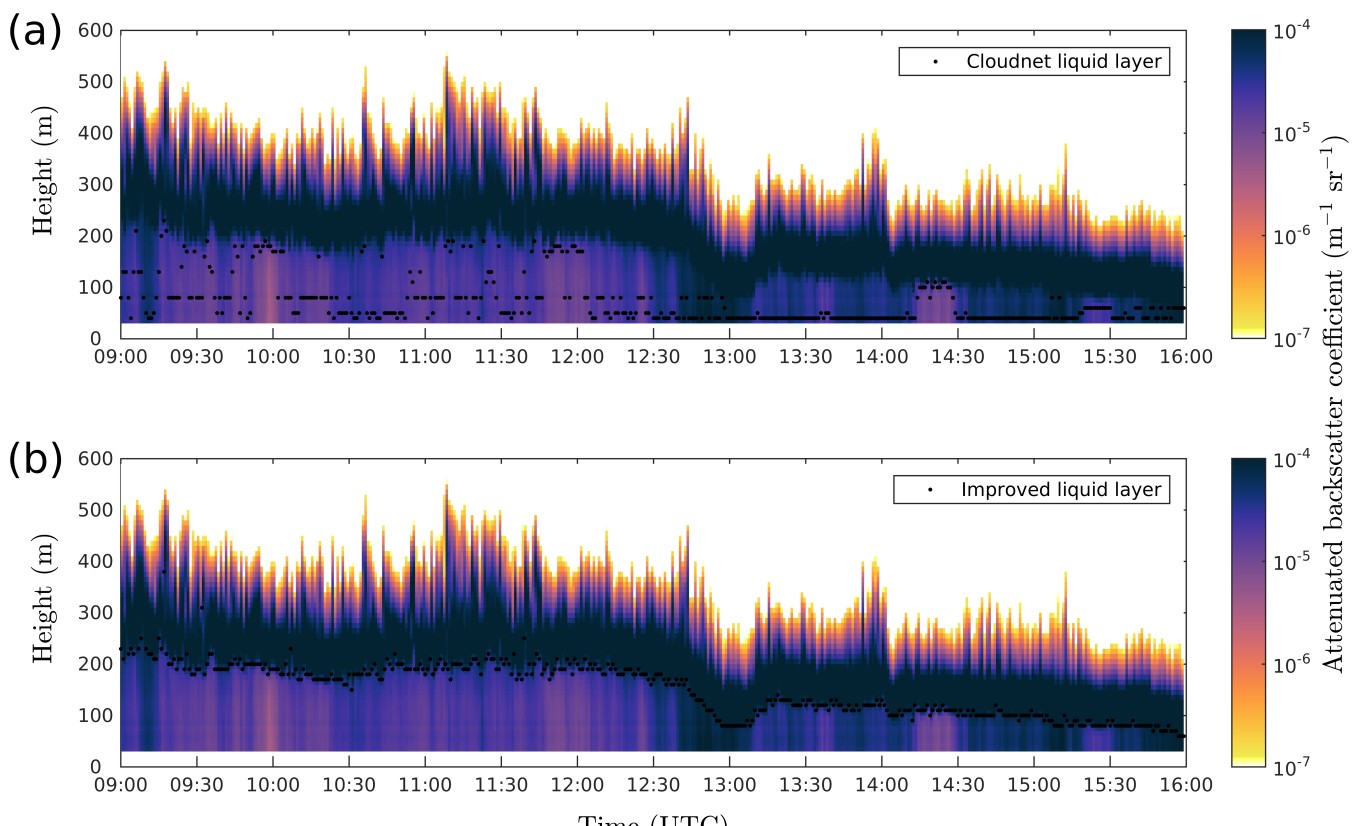

**Figure 2.** Time-height cross section of attenuated backscatter profiles from a Vaisala CL51 ceilometer on 27th October 2016 at Helsinki, Finland, with the Cloudnet approach (a) and with our updated algorithm (b) for obtaining liquid layer base. A major improvement is seen during precipitation events.



**Table 1.** ECMWF IFS model variables. Model-level fields have a vertical dimension. Single-level fields have no vertical dimension; this includes surface fields and column-integrated fields.

| Variable | Short name | Unit | Variable type | Other |
|---|---|---|---|---|
| Low cloud cover | LCC | 0–1 | single-level | instant |
| Medium cloud cover | MCC | 0–1 | single-level | instant |
| Specific cloud liquid water content | CLWC | $kg\ kg^{-1}$ | model-level | instant |
| Temperature | T | K | model-level | instant |
| Pressure | PRES | Pa | model-level | instant |
| Surface pressure | SP | Pa | single-level | instant |
| Surface solar radiation downwards | SSRD | $J\ m^{-2}$ | single-level | cumulative |
| TOA incident solar radiation | TISR | $J\ m^{-2}$ | single-level | cumulative |





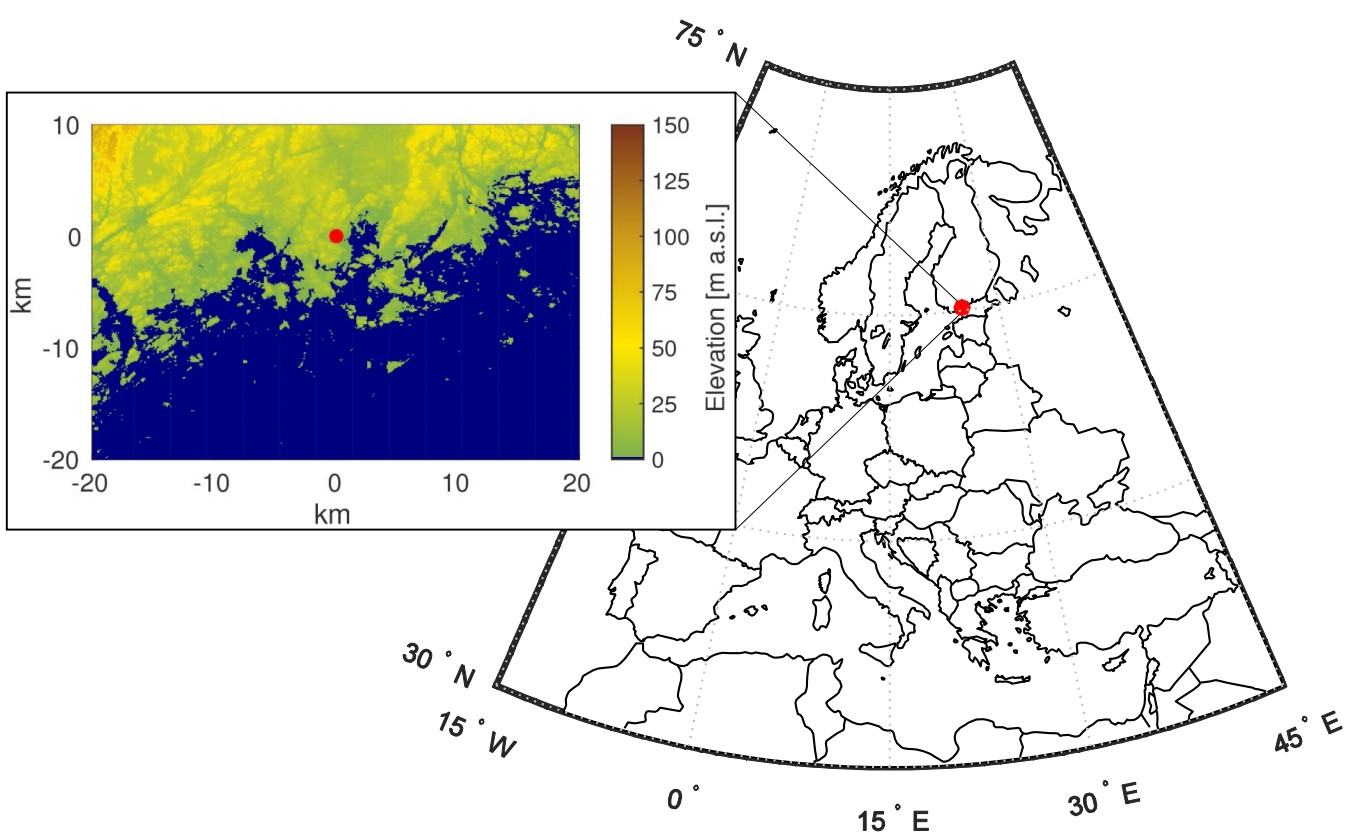

**Figure 3.** Measurement site at Helsinki, Finland (60.204° N, 24.961° E).





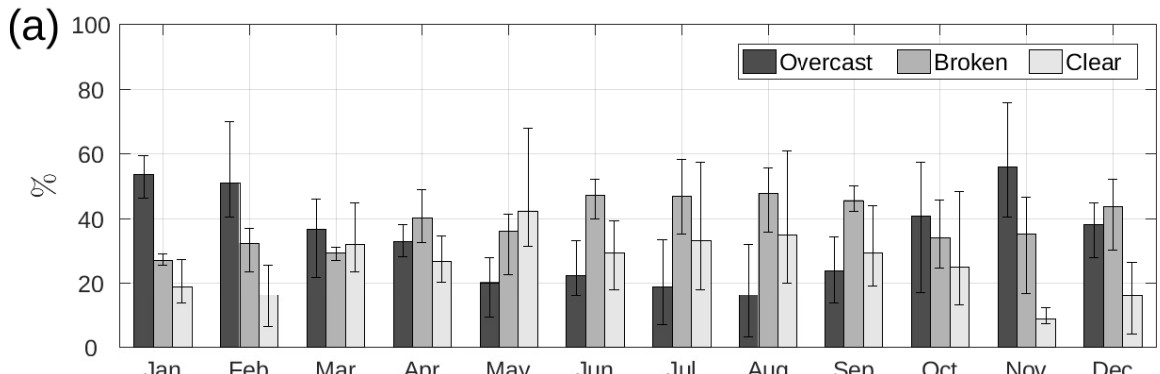

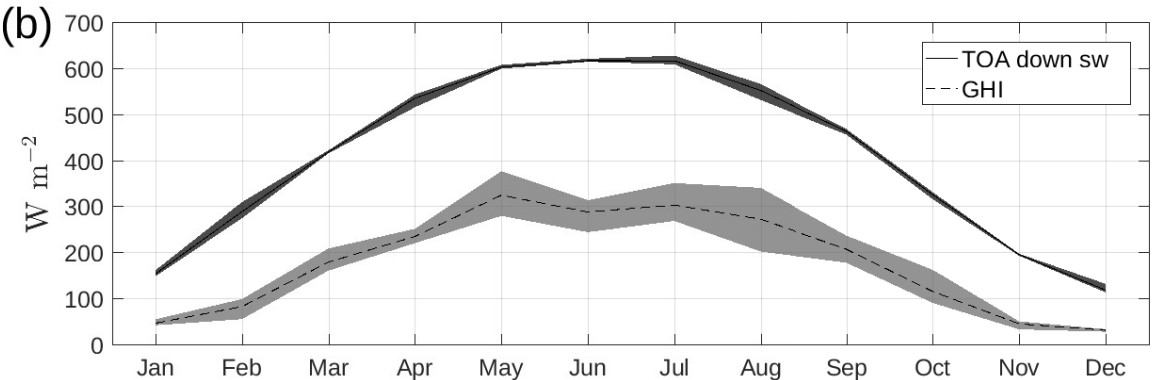

**Figure 4.** Relative occurrence of overcast, broken cloud and clear sky conditions (a). Bars show yearly variation (min, max). Annual variation in observed GHI and forecast top of the atmosphere downwelling shortwave radiation (b). Shaded area represents the year-to-year variation in monthly means.






**Figure 5.** Seasonal normalised density scatter plots of observed and forecast cloud cover (total counts for each season are given in the titles). Seasons are defined based on the annual distribution of incoming solar radiation: Winter (November to January), Spring (February to April), Summer (May to July), and Autumn (August to October).





**Figure 6.** Relative occurrence of elements in the contingency table (hit, false alarm, miss, and correct negative) for each month with a cloud cover threshold of 0.05 (a). Monthly mean skill scores for cloud cover: MAESS (b) and MSESS (c), individual monthly mean for each year (dots) and four-year average (line).





**Figure 7.** Seasonal normalised density scatter plots of observed and forecast GHI (total counts for each season are given in the titles). Seasons are defined based on the annual distribution of incoming solar radiation: Winter (November to January), Spring (February to April), Summer (May to July), and Autumn (August to October).





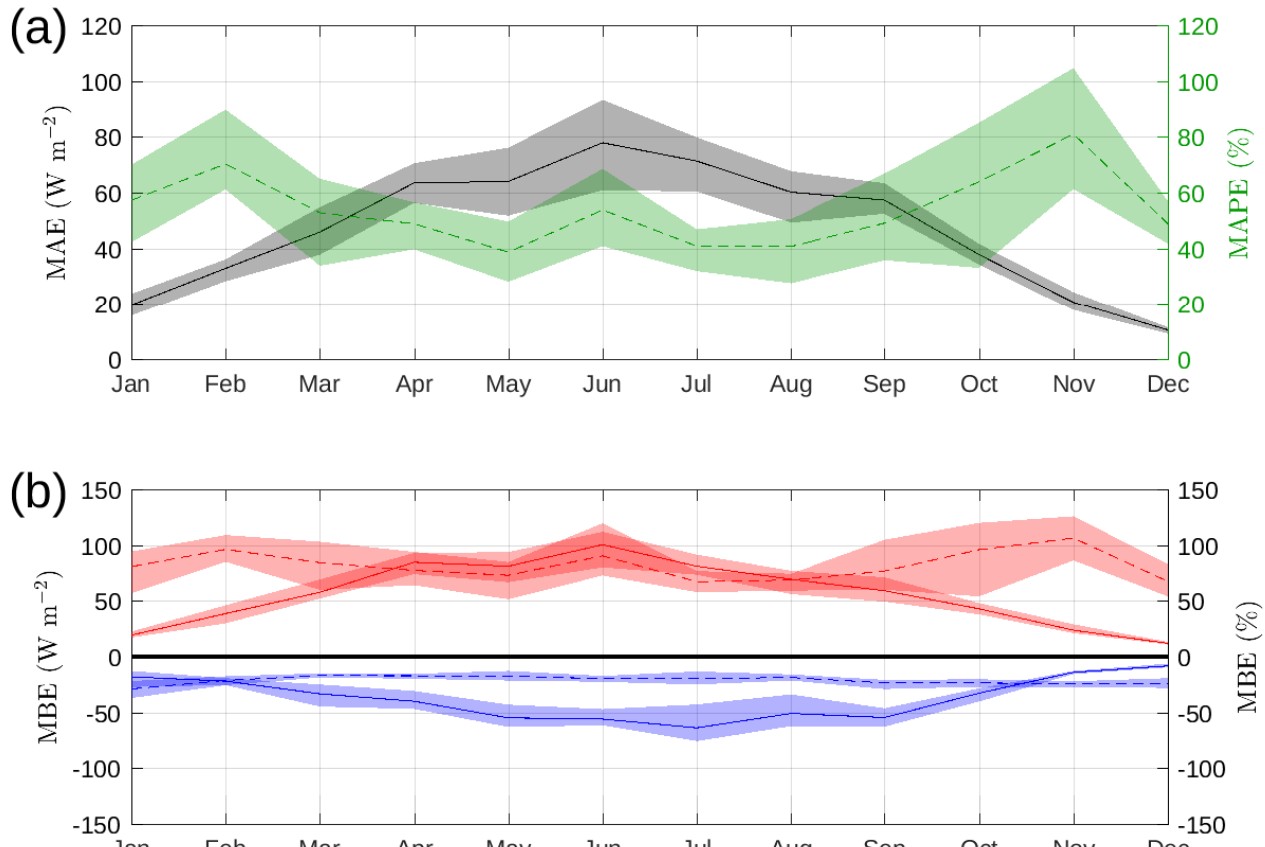

**Figure 8.** Monthly MAE (black solid line) and MAPE (green dashed line) in solar radiation forecast (a). Monthly absolute (solid line) and relative (dashed line) MBE (b). Positive bias (red) and negative bias (blue) are shown separately; shaded area represents year-to-year variation.



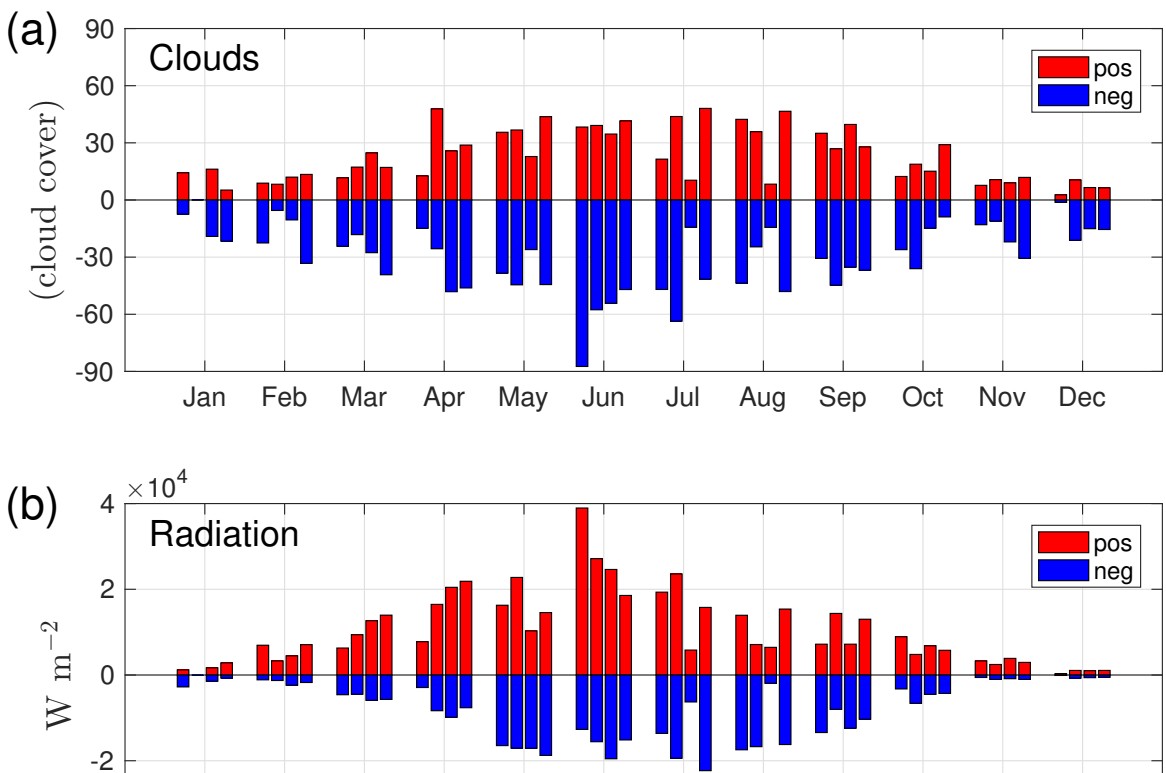

**Figure 9.** Monthly accumulated positive (red) and negative (blue) bias in cloud cover forecast (a) and solar radiation forecast (b). The four bars in each month represent individual years (2014–2017).




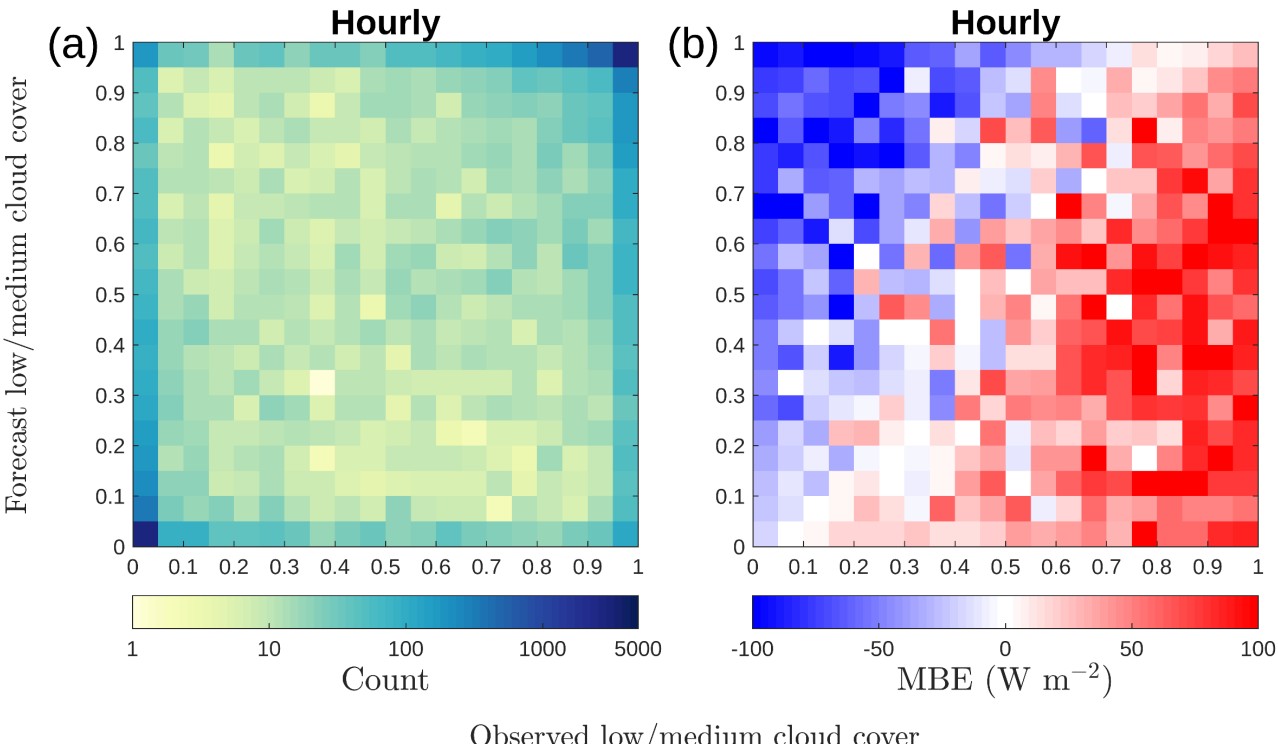

**Figure 10.** 2D-histogram of observed and forecast cloud cover (a), with colors representing counts on a logarithmic scale, and MBE in solar radiation forecast (b) for each cloud cover pair in (a).


**Figure 11.** Same plots as Fig 10, except for different averaging time-windows.



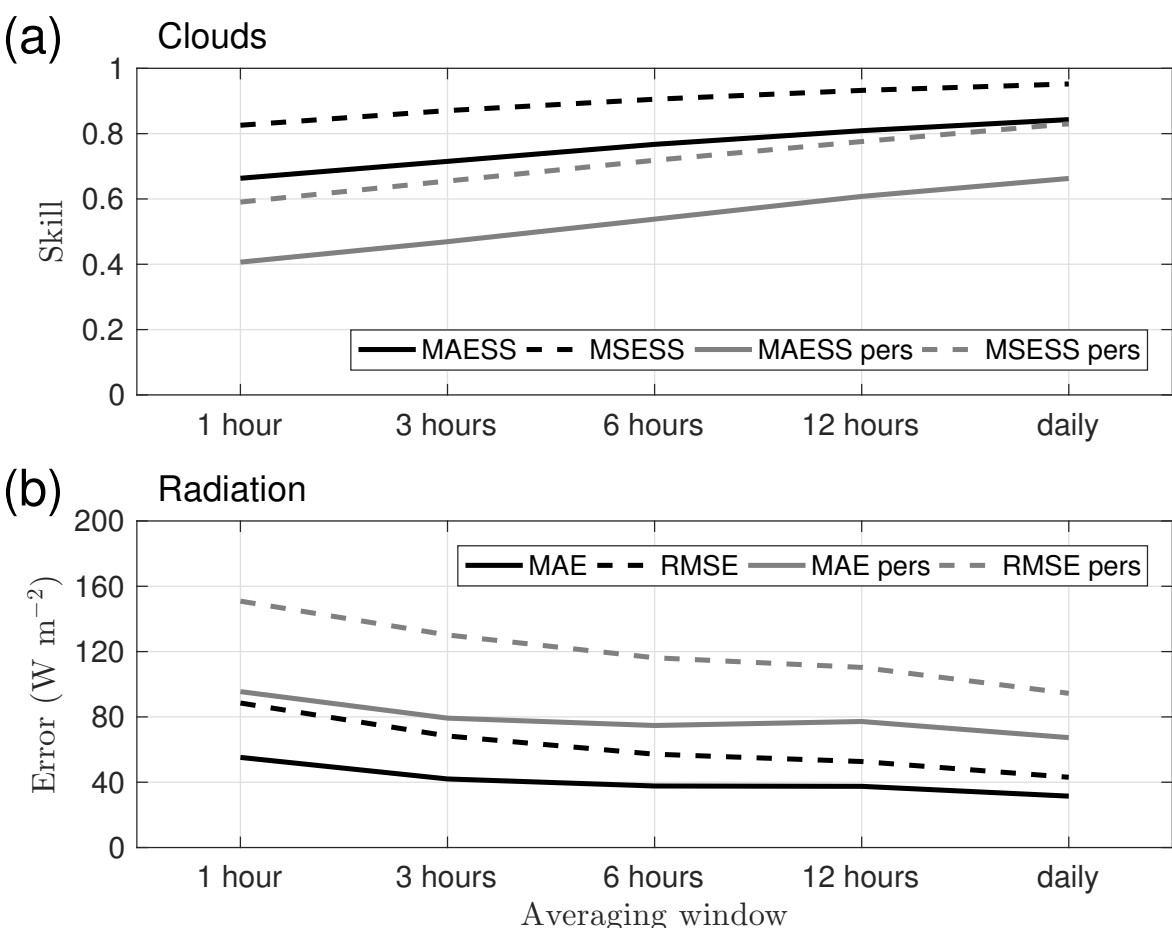

**Figure 12.** Cloud cover forecast skill scores (a) and error in solar radiation forecast (b) for different averaging time-windows, including persistence forecasts (grey lines). Note the non-linear x-axis.





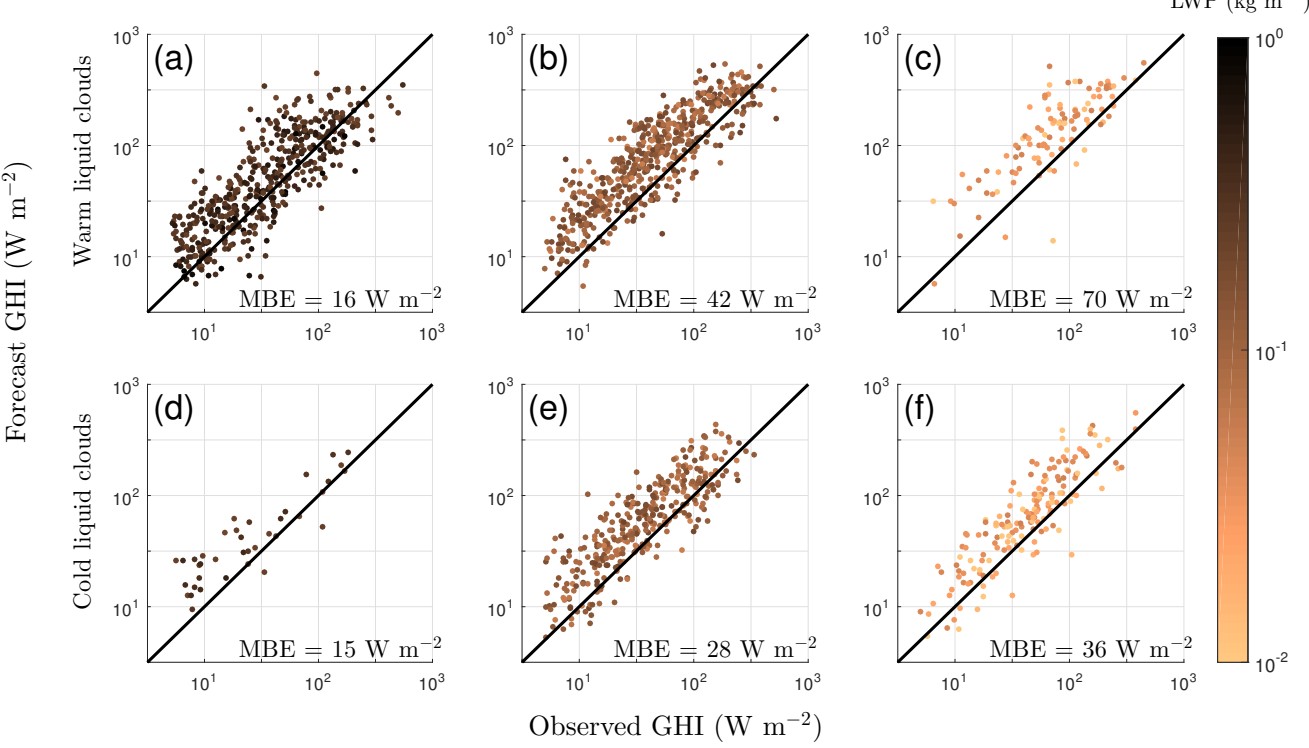

**Figure 13.** Solar radiation forecast MBE versus forecast LWP for different LWP and temperature classes: warm clouds (a)–(c) with cloud base temperature above 0 °C and cold (supercooled) clouds (d)–(f) with cloud base temperature less than 0 °C; LWP > 0.2 kg m$^{-2}$ (a), (d); 0.2 kg m$^{-2}$ ≥ LWP ≥ 0.05 kg m$^{-2}$ (b), (e); LWP < 0.05 kg m$^{-2}$ (c), (f). Color scale indicates LWP values.