# Peer review of "Evaluating solar radiation forecast uncertainty"

_Atmospheric Chemistry and Physics, 2018_

## Referee Comment (RC1) · Anonymous Referee #1 · 3 Dec 2018

This is an excellently written paper that has a clear purpose, structure and message. The figures are clear, the method nicely builds on, and quotes, previous work and the conclusions are traceable and of interest.

Really interesting to see the result that change from a - to + bias does not follow the diagonal. The discussion on what the non-zero bias at (0,0) and (1,1) implies is a nice way of getting to two key results: that there is a not enough solar radiation reaching the surface when the model correctly predicts clear sky and that there is too much when the model correctly predicts overcast conditions. This is a useful technique for identifying issues with, probably aerosols, and in-cloud water paths. Also interesting to see the result that at this location "clouds are forecast less skilfully in summer, which is when the solar resource is greatest."

[Figure]

This paper could probably be accepted as it is, but for thoroughness I include a list of typographical issues and two minor science questions:

Minor Issues:

p12, l 33, "a persistence forecast uses the forecast from the day before". Are you sure you don't mean "a persistence forecast uses the OBSERVATIONS from the day before", also I guess these are the "HOURLY observations".

p13, l 28 Could you include the formula for the regression that allows you to de-bias your data? I realise that this may only really be applicable at this location and if it were included others may be tempted to apply it elsewhere, so I understand if you would rather not.

Typography:

p2, l 10: suggest changing "by the ECMWF" to "of the ECMWF" p3, l 17: delete comma after "therefore". And remove THE in "do not use these the values". p3, l 25: Kotthaus ref place the ( after the name. p3, l 33: no need for "clearly" p8, lines 11-13 and lines 15-17, these sentences seem like a contradiction (one says you are using a sum (i.e. maximum overlap), then you say random overlap... Do lines 15-17 need to be included at all?

End of review

---

## Referee Comment (RC2) · Anonymous Referee #2 · 4 Dec 2018

**'Evaluating solar radiation forecast uncertainty', by Tuononen et al.**
Review, 4 Dec 2018

This is a very well written and structured paper presenting a comprehensive evaluation of solar radiation forecast skill of a global model for one specific location. Care has been taken to pre-process both the observations and forecasts in order to reduce representativeness issues. The methodology presented allows to draw conclusions about the nature of model deficiencies, and could be applied in other locations. The manuscript is basically ready for publication, below are just a few minor comments.

*Minor comments*

Page 1, line 18: Temporal averaging cannot have an effect on the overall bias. After reading the paper, I assume what the authors mean is that the temporal averaging had little impact on the magnitude of the positive and negative contributions to the overall bias.

Page 6, line 14: 'updated .. to the computationally cheaper ECRAD scheme ..' The new scheme was cheaper but also contained scientific developments which slightly improved forecast skill, according to Hogan and Bozzo (2016).

Page 9, line 1: The bias, or mean error, is usually abbreviated 'ME' (see e.g. Wilks, 1995). To keep with this convention, I would replace 'Mean Biased Error' by 'Mean Error' and 'MBE' by 'ME'.

Page 11, lines 16-17: There appears to be a repetition here. In line 16 '..the relative negative MBE is rather constant around 25%.' And in line 17 'Negative relative MBE is constant throughout the year , ..'

Page 11, line 21: The statement '.. only the forecast of cloud impacting the solar radiation forecast ..' is not quite correct in this context, since aerosol and/or humidity content could be wrong in the model, which could lead to radiation errors even with a perfect radiative transfer model.

Page 12, line 26: Temporal averaging cannot have an effect on the overall bias.

*Typological*

Page 3, line 17: 'we do not use these values.'

Page 3, line 25: 'recommended by Kotthaus et al. (2016).'

Page 10, line 6: 'For a perfect forecast, all values'

Page 12, line 16: 'occur more frequently than clear sky' (to make it clearer)

Page 12, lines 22 and 31: 'number of .. false alarms .. decreases' and 'and error decreasing'

Page 13, line 22: 'They found a positive radiation bias'

Page 14, line 26: 'the source of the positive bias'

---

## Referee Comment (RC3) · Anonymous Referee #3 · 6 Dec 2018

**6 Dec 2018**

*General comments*

The manuscript prepared by Tuononen et al. evaluates the surface downwelling solar irradiance from the IFS model by comparing 4 years of observations from one location in Helsinki, Finland, with model output at the nearest grid point. Overall, the model bias in the surface solar irradiance is positive. This positive bias results from a combination of negative biases in less-frequent clear-sky conditions and positive biases in more-frequent overcast conditions. As part of the analysis, a new algorithm is also presented for improved detection of cloud base, precipitation and fog from ceilometer observations, which can be applied at other sites.

The paper is very nicely written, with clear motivation and aims, well thought out methods, and concise results. The paper is almost ready and I recommend publication after addressing the minor comments outlined below.

*Specific comments*

P6, L21: "obtained from the closest land grid point to the measurement site". How close is this exactly? And how much does this distance change when the resolution of the model increased from 16 to 9 km? I think numbers should be mentioned here.

P7, L25–26: "one hour averaging corresponds to advection speeds of 4.5 m s $^{-1}$ or 2.5 m s$^{-1}$". I generally like the idea to use temporal averaging of the observations to better match the spatial scale of the model, but I think this could have been handled better. Specifically, I think the analysis would have been more consistent if observed (or even modelled) wind speeds were used to define the appropriate averaging time of the observations on a case-by-case basis. I do not suggest the authors change their analysis, but they should provide a sentence or two to support their decision. For example, are the corresponding advection speeds of 4.5 m s $^{-1}$ and 2.5 m s$^{-1}$ at least close to climatological wind speeds at this site?

P8, L2–3: "Additionally, a cloud base may not be detected in strong precipitation due to the attenuation of the lidar signal". I found this statement a bit contradictory to the earlier results presented in Fig. 2. Perhaps it could be rephrased or, if this is now an infrequent issue, it could be left out to avoid confusion.

P8, L16–17: "which may result in a slight overestimate". Seems a bit vague. Could a reference be provided here?

P8, L18: "5.2". For the comparison of surface shortwave irradiance between model and observation, I think one important difference has been overlooked. The observations see the entire hemisphere above the given location and are therefore inherently 3D. In contrast, the model output is likely a result of 1D radiative transfer, using only the atmospheric properties of the vertical column at the given location. Under homogeneous conditions (eg. clear-sky or overcast), this may not be important. But Fig. 4 shows the prevalence of broken

cloud in summer for which 3D effects can be large. I think the authors need to acknowledge that they are aware of this difference (3D vs. 1D), even if they are not able to account for it.

P9, L19: "Thus, the amount of solar radiation at the top of the atmosphere is much higher during summer". Not just because the length of the day is longer in summer, but also because the sun reaches higher in the sky (will scale as the cosine of the solar zenith angle).

P9, L22: "clouds and the atmosphere". Better to mention aerosols explicitly here. Perhaps "clouds, aerosols and atmospheric gases".

*Technical corrections*

P3, L16: "cloud contain" -> "cloud contains"

P3, L25: "(Kotthaus et al., 2016)" -> "Kotthaus et al. (2016)"

P6, L20: "corresponding" -> "correspond"

P6, L26: "(LCC; Table 1) … (MCC)". Seems inconsistent, should probably cite Table 1 in both brackets, or not at all.

P8, L27-28: "therefore penalizing larger errors more than small but more common differences". This doesn't make sense to me, consider re-phrasing.

P10, L6: "forecasts" -> "forecast"

P11, L24: "cloud radiative properties" -> "cloud radiative effect"

P13, L29: "to remove the observed bias" -> "to remove the bias"

---

## Author Comment (AC1) · 29 Dec 2018

We have addressed all of the points raised by the reviewer (copied here and shown in red text), and include our responses to each point below (in black text).

**Reviewer 1**

This is an excellently written paper that has a clear purpose, structure and message. The figures are clear, the method nicely builds on, and quotes, previous work and the conclusions are traceable and of interest. Really interesting to see the result that change from a - to + bias does not follow the diagonal. The discussion on what the non-zero bias at (0,0) and (1,1) implies is a nice way of getting to two key results: that there is a not enough solar radiation reaching the surface when the model correctly predicts clear sky and that there is too much when the model correctly predicts overcast conditions. This is a useful technique for identifying issues with, probably aerosols, and in-cloud water paths. Also interesting to see the result that at this location "clouds are forecast less skilfully in summer, which is when the solar resource is greatest. This paper could probably be accepted as it is, but for thoroughness I include a list of typographical issues and two minor science questions:"

We thank the reviewer for their comments on our submitted manuscript "Evaluating solar radiation forecast uncertainty". Based on the comments and suggestions by the reviewer, we have revised our manuscript.

Minor Issues:

p12, l 33, "a persistence forecast uses the forecast from the day before". Are you sure you don't mean "a persistence forecast uses the OBSERVATIONS from the day before", also I guess these are the "HOURLY observations".

We use hourly forecast values as our persistence forecast. Therefore, we just simply keep the same forecast as was forecast to the previous day to represent the "persistent condition". We updated the sentence to state: "a persistence forecast uses the hourly forecast values from the day before."

p13, l 28 Could you include the formula for the regression that allows you to de-bias your data? I realise that this may only really be applicable at this location and if it were included others may be tempted to apply it elsewhere, so I understand if you would rather not.

As pointed out by the reviewer, our aim in this study was to develop methods applicable to any site. At this particular site, we noted that the relative bias appeared to be constant across a wide range of GHI values, hence the possibility of bias correction. However, this constant bias may not be true at other locations, and would require

further analysis. We decided not to include this bias correction in the manuscript.

Typography:
p2, l 10: suggest changing "by the ECMWF" to "of the ECMWF"
Changed.

p3, l 17: delete comma after "therefore". And remove THE in "do not use these the values".
Changed.

p3, l 25: Kotthaus ref place the ( after the name.
Corrected.

p3, l 33: no need for "clearly"
Removed.

p8, lines 11-13 and lines 15-17, these sentences seem like a contradiction (one says you are using a sum (i.e. maximum overlap), then you say random overlap... Do lines 15-17 need to be included at all?
Lines 6–14 refer to the observations, where we explain how the observed cloud cover values are treated. Lines 15-17 refer to the model forecasts. Observations of lcc and mcc are summed due to the nature of observations (time series of cloud hits), whereas we need to combine forecast lcc and mcc.

---

## Author Comment (AC2) · 29 Dec 2018

We have addressed all of the points raised by the reviewer (copied here and shown in red text), and include our responses to each point below (in black text).

**Reviewer 2**

This is a very well written and structured paper presenting a comprehensive evaluation of solar radiation forecast skill of a global model for one specific location. Care has been taken to pre-process both the observations and forecasts in order to reduce representativeness issues. The methodology presented allows to draw conclusions about the nature of model deficiencies, and could be applied in other locations. The manuscript is basically ready for publication, below are just a few minor comments. We thank the reviewer for their comments on our submitted manuscript "Evaluating solar radiation forecast uncertainty". Based on the comments and suggestions by the reviewer, we have revised our manuscript.

Minor comments:
Page 1, line 18: Temporal averaging cannot have an effect on the overall bias. After reading the paper, I assume what the authors mean is that the temporal averaging had little impact on the magnitude of the positive and negative contributions to the overall bias.
Yes, we agree with the reviewer that we did not word this very well. We have revised the sentence to say: "Temporal averaging improved the cloud cover forecast and hence decreased the solar radiation forecast error."

Page 6, line 14: 'updated..to the computationally cheaper ECRAD scheme ..' The new scheme was cheaper but also contained scientific developments which slightly improved forecast skill, according to Hogan and Bozzo (2016).
We did not want to give the impression that we neglected the scientific improvements made to the radiation scheme. We have revised the text to say: "Notably, the radiation scheme was updated from McRad scheme (Morcrette et al., 2008) to the scientifically improved and computationally cheaper ECRAD scheme (Hogan and Bozzo, 2016) in 2016."

Page 9, line 1: The bias, or mean error, is usually abbreviated 'ME' (see e.g. Wilks, 1995). To keep with this convention, I would replace 'Mean Biased Error' by 'Mean Error' and 'MBE' by 'ME'.
We have made this change throughout the manuscript.

Page 11, lines 16-17: There appears to be a repetition here. In line 16 '..the relative negative MBE is rather constant around 25 %.' And in line 17 'Negative relative MBE is constant throughout the year, ..'

We have removed the repetition from line 17.

Page 11, line 21: The statement '..only the forecast of cloud impacting the solar radiation forecast..' is not quite correct in this context, since aerosol and/or humidity content could be wrong in the model, which could lead to radiation errors even with a perfect radiative transfer model.

Here, we just wanted to remind readers that increasing the forecast cloud cover should reduce the forecast of solar radiation reaching the surface. Our sentence was supposed to imply that the profiles of humidity and aerosols were correct; even if they are not correct, as long as they do not change, the statement is still valid. We revised this sentence: "Assuming the correct representation of radiative transfer in the atmosphere, with only the forecast of cloud impacting the solar radiation forecast at the surface (no change in aerosol or humidity), then an increase in forecast cloud cover would be expected to result in a reduction in the amount of forecast solar radiation."

Page 12, line 26: Temporal averaging cannot have an effect on the overall bias.

We agree that overall bias should not change with temporal averaging, however, we noted that there were some slight differences due to our conditional sampling (how we handle occasional gaps in observations) changing slightly for the different temporal averaging periods. We have checked this and revised the statement: "The overall bias remains around 8 W m$^{-2}$."

Typological:

All typological points below are corrected in the text.

Page 3, line 17: 'we do not use these values.'
Page 3, line 25: 'recommended by Kotthaus et al. (2016).'
Page 10, line 6: 'For a perfect forecast, all values'
Page 12, line 16: 'occur more frequently than clear sky' (to make it clearer)
Page 12, lines 22 and 31: 'number of .. false alarms .. decreases' and 'and error decreasing'
Page 13, line 22: 'They found a positive radiation bias'
Page 14, line 26: 'the source of the positive bias'

---

## Author Comment (AC3) · 29 Dec 2018

We have addressed all of the points raised by the reviewer (copied here and shown in red text), and include our responses to each point below (in black text).

**Reviewer 3**

General comments
The manuscript prepared by Tuononen et al. evaluates the surface downwelling solar irradiance from the IFS model by comparing 4 years of observations from one location in Helsinki, Finland, with model output at the nearest grid point. Overall, the model bias in the surface solar irradiance is positive. This positive bias results from a combination of negative biases in less-frequent clear-sky conditions and positive biases in more-frequent overcast conditions. As part of the analysis, a new algorithm is also presented for improved detection of cloud base, precipitation and fog from ceilometer observations, which can be applied at other sites. The paper is very nicely written, with clear motivation and aims, well thought out methods, and concise results. The paper is almost ready and I recommend publication after addressing the minor comments outlined below.

We thank the reviewer for their comments on our submitted manuscript "Evaluating solar radiation forecast uncertainty". Based on the comments and suggestions by the reviewer, we have revised our manuscript.

Specific comments
P6, L21: "obtained from the closest land grid point to the measurement site". How close is this exactly? And how much does this distance change when the resolution of the model increased from 16 to 9 km? I think numbers should be mentioned here.

We use the Meteorological Archival and Retrieval System (MARS) at ECMWF to obtain the data and we were advised by ECMWF to use a grid resolution of 0.125° in the retrieval API used for downloading data. This means that the data we receive is transformed from the internal model representation (spherical harmonic) and resolution, to a regular lat/lon grid. Therefore, our retrieved model grid point has the same lat/lon before and after the model resolution change. We have modified the text to describe this, here: ".. obtained from the closest land grid point to the measurement site, 2.1 km away."; and in Table 1: "Obtained via the Meteorological Archival and Retrieval System (MARS) at ECMWF using a grid resolution of 0.125°."

P7, L25–26: "one hour averaging corresponds to advection speeds of 4.5 m s -1 or 2.5 m s-1". I generally like the idea to use temporal averaging of the observations to better match the spatial scale of the model, but I think this could have been handled

better. Specifically, I think the analysis would have been more consistent if observed (or even modelled) wind speeds were used to define the appropriate averaging time of the observations on a case-by-case basis. I do not suggest the authors change their analysis, but they should provide a sentence or two to support their decision. For example, are the corresponding advection speeds of 4.5 m s$^{-1}$ and 2.5 m s$^{-1}$ at least close to climatological wind speeds at this site?

Yes, we agree that spatial averaging based on the wind speed at the cloud level would be ideal. However, this would require the download of wind and cloud fields on model levels; we take the single-level cloud forecasts as we were developing a simple and robust methodology which can be applied rapidly to numerous sites globally. Hence, we used one-hour averaging, which has been used by many other researchers. We suspect that one-hour averaging is substantially longer than required to meet the advective-averaging time scale, which likely corresponds to advection speeds above 10 m s$^{-1}$ at the cloud altitudes at this location. We do mention in the text that care should be taken.

P8, L2–3: "Additionally, a cloud base may not be detected in strong precipitation due to the attenuation of the lidar signal". I found this statement a bit contradictory to the earlier results presented in Fig. 2. Perhaps it could be rephrased or, if this is now an infrequent issue, it could be left out to avoid confusion.

Heavy precipitation may be a frequent occurrence in some locations so the possibility of this situation occurring should always be kept in mind. Note that Fig. 2 does not represent heavy precipitation. We have rephrased this sentence: "In strong precipitation, the lidar signal may be sufficiently attenuated so that the liquid cloud base can no longer be detected."

P8, L16–17: "which may result in a slight overestimate". Seems a bit vague. Could a reference be provided here?

We have added a reference:

Hogan, R. J. and Illingworth, A. J. (2000), Deriving cloud overlap statistics from radar. Q.J.R. Meteorol. Soc., 126: 2903-2909. doi:10.1002/qj.49712656914

P8, L18: "5.2". For the comparison of surface shortwave irradiance between model and observation, I think one important difference has been overlooked. The observations see the entire hemisphere above the given location and are therefore inherently 3D. In contrast, the model output is likely a result of 1D radiative transfer, using only the atmospheric properties of the vertical column at the given location. Under homogeneous conditions (eg. clear-sky or overcast), this may not be important. But

Fig. 4 shows the prevalence of broken cloud in summer for which 3D effects can be large. I think the authors need to acknowledge that they are aware of this difference (3D vs. 1D), even if they are not able to account for it.
Thanks for this comment. Yes, we are aware that the model may not include all 3D radiative transfer effects. We have added a sentence in section 5.2 to acknowledge this issue:"It should be noted that the model radiative transfer scheme is unlikely to completely account for the 3-dimensional nature of radiative transfer as experienced by the observations."

P9, L19: "Thus, the amount of solar radiation at the top of the atmosphere is much higher during summer". Not just because the length of the day is longer in summer, but also because the sun reaches higher in the sky (will scale as the cosine of the solar zenith angle).
We have revised the text to say: "Due to the change in the solar zenith angle, the length of the shortest day of the year (winter solstice on 21st or 22nd December) is less than 6 hours and the length of the longest day (summer solstice between 20th and 22nd June) is almost 19 hours. The amount of solar radiation at the top of the atmosphere is much higher during summer when the solar zenith angle is also much higher (Fig. 4b, solid line)."

P9, L22: "clouds and the atmosphere". Better to mention aerosols explicitly here. Perhaps "clouds, aerosols and atmospheric gases".
We made this change.

Technical corrections
P3, L16: "cloud contain" -> "cloud contains"
Corrected.

P3, L25: "(Kotthaus et al., 2016)" -> "Kotthaus et al. (2016)"
Corrected.

P6, L20: "corresponding" -> "correspond"
Corrected.

P6, L26: "(LCC; Table 1) ... (MCC)". Seems inconsistent, should probably cite Table 1 in both brackets or not at all.
We have removed the reference to Table 1, as we already stated "A list of the model variables we use is given in Table 1" in line 22.

P8, L27–28: "therefore penalizing larger errors more than small but more common differences". This doesn't make sense to me, consider re-phrasing.
We have revised this sentence: "MAESS uses the absolute difference between forecast and observed value, and MSESS uses the squared difference, which for two forecasts with the same absolute error, will penalize the forecast with one or two large errors more than the forecast with many small errors."

P10, L6: "forecasts" -> "forecast"
Corrected.

P11, L24: "cloud radiative properties" -> "cloud radiative effect"
Changed.

P13, L29: "to remove the observed bias" -> "to remove the bias"
Changed.